# Subregions in the ventromedial prefrontal cortex integrate threat and protective information to meta-represent safety

Sarah M. Tashjian[1,2]*, Joseph Cussen[1], Wenning Deng[2], Bo Zhang[2], Dean Mobbs[2,3]

1 School of Psychological Sciences, University of Melbourne, Parkville, Australia, 2 Humanities and Social Sciences, California Institute of Technology, Pasadena, California, United States of America, 3 Computation and Neural Systems, California Institute of Technology, Pasadena, California, United States of America

* sarah.tashjian@unimelb.edu.au

**Data Availability Statement:** Task code and behavioral data are available through OSF, https://osf.io/hw3r9, DOI 10.17605/OSF.IO/8QG7Y. Neuroimaging data are available through Science

## Abstract

Pivotal to self-preservation is the ability to identify when we are safe and when we are in danger. Previous studies have focused on safety estimations based on the features of external threats and do not consider how the brain integrates other key factors, including estimates about our ability to protect ourselves. Here, we examine the neural systems underlying the online dynamic encoding of safety. The current preregistered study used 2 novel tasks to test 4 facets of safety estimation: *Safety Prediction*, *Meta-representation*, *Recognition*, and *Value Updating*. We experimentally manipulated safety estimation changing both levels of external threats and self-protection. Data were collected in 2 independent samples (behavioral $N = 100$; MRI $N = 30$). We found consistent evidence of subjective changes in the sensitivity to safety conferred through protection. Neural responses in the ventromedial prefrontal cortex (vmPFC) tracked increases in safety during all safety estimation facets, with specific tuning to protection. Further, informational connectivity analyses revealed distinct hubs of safety coding in the posterior and anterior vmPFC for external threats and protection, respectively. These findings reveal a central role of the vmPFC for coding safety.

## Introduction

In their natural habitat, the Sundarbans Tiger, which is a formidable threat to humans, is justifiably feared. However, with a gun in hand, we fear the Tiger less. At the zoo, behind the protection of laminated glass, fear is replaced with enthused curiosity. In all of these scenarios, the sensory features of the Tiger remain stable, yet the perception of safety fluctuates. These fluctuations in safety occur as a function of information unrelated to the Tiger's features, but changes in the perception of safety. Despite this knowledge, the contributions of safety estimation beyond external threats are largely ignored in the existing literature. To accurately estimate safety, we need to integrate threat-related information with information about our ability to protect ourselves [1,2]. From an evolutionary perspective, these factors determine how likely we are to succeed in surviving encounters with natural dangers.

Data Bank, https://www.scidb.cn/en/s/BNjaUz, DOI 10.17605/OSF.IO/8QG7Y.

**Funding:** The study was supported by the following grants to SMT: US National Science Foundation Directorate for Social, Behavioral and Economic Sciences Postdoctoral Research Fellowship 2203522 (https://www.nsf.gov/); Brain and Behavior Research Foundation Young Investigator Grant 30788 (https://bbrfoundation.org/). The study was supported by the following grants to DM: US National Institute of Mental Health grant R01MH133730 (https://www.nimh.nih.gov/); John Templeton Foundation grant TWCF0366 (https://www.templeton.org/). The funders played no role in study design, data collection or analysis, decision to publish, or preparation of the manuscript.

**Competing interests:** The authors have declared that no competing interests exist.

**Abbreviations:** ACC, anterior cingulate cortex; CSF, cerebrospinal fluid; DMN, default mode network; EPI, echo-planar imaging; FD, framewise displacement; fMRI, functional magnetic resonance imaging; FWE, familywise error; GM, gray-matter; INU, intensity non-uniformity; ISI, interstimulus interval; ITI, inter-trial interval; ROI, region of interest; rTMS, repetitive transcranial magnetic stimulation; SD, standard deviation; TFCE, threshold-free cluster enhancement; vmPFC, ventromedial prefrontal cortex; WM, white-matter.

The ability to recognize safety is critical for adjusting adaptive defensive responses, reducing stress, and initiating other survival behaviors, including foraging and mating [3,4]. How the brain integrates multiple sources of dynamic information to compute safety estimates remains to be identified. One theory is that the coding of safety is constructed based on representations that reflect the learned and phenotypic features of external threats [5–7]. The brain creates an internal model that integrates incoming sensory information about a stimulus with other relevant information such as context (i.e., Am I at the zoo or alone in the forest?). This "other" information does not pertain to the threat itself but is important for accurately estimating safety. As relevant information changes so can the safety estimate, even if the threat itself remains unchanged. Similarly, changing threat features (i.e., Is the tiger awake or asleep?) should trigger updated safety estimates through integration in safety circuits with consequences for survival behavior. Such *Meta-representations* are likely to involve neuronal ensembles that integrate the valuation of threats with information about our ability to protect against them [8,9]. Depending on the resulting calculation, defensive excitation or inhibition occurs [10–13].

We test 2 primary hypotheses regarding the functioning of safety neural circuitry. First, we test the hypothesis that the neural systems involved in representing threat and protection are dissociable during *Safety Prediction* [2]. We extend beyond models focused on the external environment to examine fluctuations in safety as they relate to self-relevant states (e.g., the value of protection). We argue that self-relevant states—the extent to which one can successfully protect oneself—are a crucial, but overlooked, factor in estimating safety. Going face-to-face with a tiger with no weapon in hand will likely result in death. Yet, with a knife, we have an increased chance of deterring the predator and surviving. A gun further increases survival likelihood and turns the tables on the tiger as the likely casualty. Second, we hypothesize that the brain integrates threat and protective information to confer a safety "*meta-representation*" [14–16]. To test this, we manipulated the safety value of threat and protective stimuli as a function of their pairing without altering the perceptual features of either stimulus. For example, a tiger becomes less dangerous when faced with a gun as opposed to a knife, but the tiger itself does not change. We hypothesize that the safety modulator (e.g., the gun or knife) will be *meta-represented* during the evaluation of the modulated stimulus (e.g., the tiger), even though the modulator is not being displayed and therefore not visually perceived.

We hypothesize a candidate region for human safety coding is the ventromedial prefrontal cortex (vmPFC). Recent theoretical developments suggest representations of threat and protective information are encoded in canonical defensive and cognitive neural circuits, the latter including the vmPFC [2,17–22]. The vmPFC aids the affective processing of safety signals as well as the acquisition of new threat associations and threat extinction during Pavlovian threat conditioning [2,17–28]. With relevance to the current study, we propose that external threats are computed in a bottom-up fashion, primarily driven by sensory processing regions of the brain, whereas protection is integrated with threat information through top-down metacognitive circuitry related to self-evaluation, including the vmPFC [29–31]. Work outside of the threat context points to the vmPFC as pivotal in supporting decisions related to the self [16] and in showing bias toward information concerning the self [31]. The vmPFC contributes to binding self-relevant information across cognitive domains such as perception, memory, and decision-making, thereby enhancing the integration of stimuli with self-representations [32]. Research on threat controllability provides links between self-referential cognition and safety —controllability, which relies on self-referential processing, improves fear extinction, which reflects safety, in humans, thus implying a role of the self in safety estimation [17–28,33]. Consequently, we identify the vmPFC is a potentially important region for integrating self and external information to formulate safety estimations.

Although we propose that successful safety estimation in humans relies on integrating multiple distinct components, how the brain interprets and weighs these factors remains unknown. No prior study to our knowledge has systematically manipulated both threat and protection concerning safety judgments. In 2 preregistered samples using a novel Safety Estimation Task (Fig 1A) (*N* = 100 behavioral; *N* = 30 functional magnetic resonance imaging, fMRI; https://osf.io/hw3r9), we examined how subjects evaluated different types of safety information (*Safety Prediction*), as well as how they integrated information to estimate safety when perceptually identical stimuli changed in value (*Safety Meta-representation*). As a

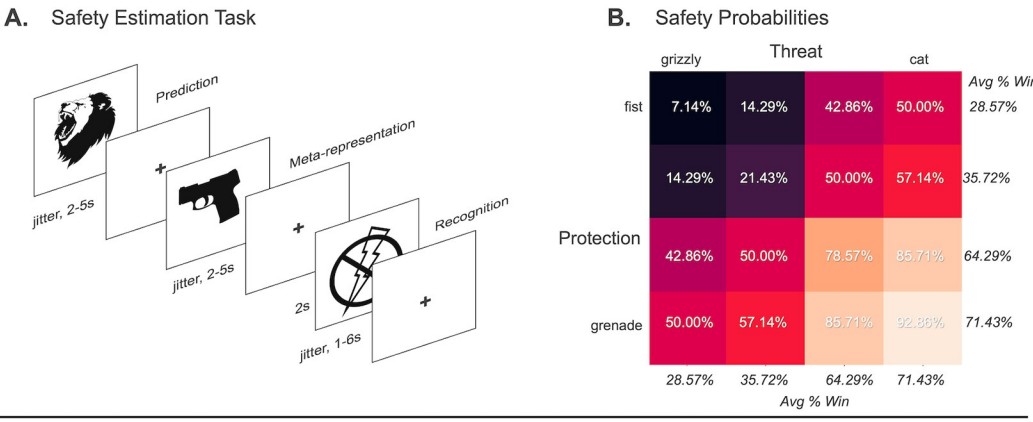

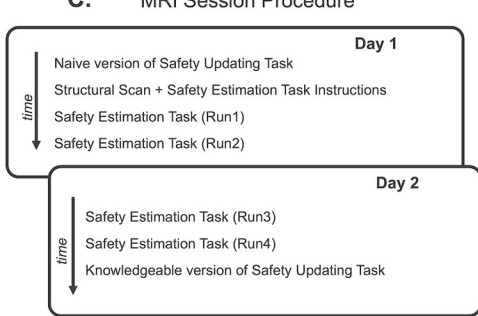

**Fig 1. Safety Estimation Task and Safety Value Updating Task schematics, including safety probabilities, a conceptual model of safety estimation and integration, and MRI session procedure details.** (**A**) **Safety Estimation Task.** An example trial is presented. Subjects were told to imagine they were battling dangerous animals with powerful weapons. On each trial, subjects saw stimuli pairs comprised of a threat (animal) and protection (weapon) with presentation of threat/protection counterbalanced. First a weapon or animal was presented (*Safety Prediction*) and subjects made an initial estimation of whether they would win or lose the battle, responding with a button press. Then, the paired stimulus was presented (*Safety Meta-representation*) and subjects made an updated judgment as to whether they would win or lose, responding with a button press. After both stimuli were presented, subjects saw the outcome of the battle depicted as either a shock (loss) or no shock (win) (*Safety Recognition*). If subjects lost and received the shock image, they had a 20% chance of receiving an electric shock to the wrist. (**B**) **Safety probabilities.** Threat and protection stimuli were each set on a four-point continuum with equivalent experimentally established safety probabilities. Italics depict the average shock value for each stimulus across all pairings. Paired probabilities are depicted in the heatmap. Probabilities were experimentally established prior to testing and were not made known to participants although the continua were easily identified based on prior knowledge and outcomes during the task. (**C**) **MRI session procedure.** On the first day, subjects completed the naive version of the Safety Value Updating Task. At this point, subjects had not completed instructions for the Safety Estimation Task and thus had no knowledge of the task or stimuli relevance. Subjects then completed a structural scan, during which they learned about the Safety Estimation Task. After completing 5 practice trials, subjects completed Run 1 and Run 2 of the Safety Estimation Task. An average of 26 min elapsed from the start of Run 1 to the start of Run 2. On the second day, subjects completed Run 3 and Run 4 of the Safety Estimation Task, with an average of 25 min from the start of Run 3 to the start of Run 4. After Run 4 ended, subjects completed the knowledgeable version of the Safety Value Updating Task. Day 2 of testing took place on average 1 day and 7 h after day 1. Task images are approximate reproductions. Source data can be found at https://osf.io/8qg7y/ under "Behavioral data".

comparison, we examined neural activation when safety was certain during the outcome phase (*Safety Recognition*). We also tested how safety estimation changed as a function of experience (*Safety Value Updating*) in a separate task administered before and after the Safety Estimation Task. We examined safety coding at the whole-brain level but focused on the contributions of the vmPFC as a hypothesized safety coding hub.

During the Safety Estimation Task, subjects were shown stimuli pairs comprised of an external threat (dangerous animal) and a self-relevant protection (powerful weapon). Four stimuli for each threat and protection were used (Fig 1B). Presentation of stimuli was counterbalanced such that in some trials, the protection was shown first and in other trials, the protection was shown second. Subjects made binary forced-choice judgments about whether they thought they would win or lose the battle against the dangerous animal using the weapon they were provided. All trials included a threat and protection, separately presented to allow for analyzing subject response when shown the first stimulus (*Safety Prediction*) separate from the second stimulus (*Safety Meta-representation*). Safety probabilities for threat and protection stimuli were matched such that high safe protection and high safe threat were equally likely to result in a win. Combinations of threat and protection stimuli were also matched with safety outcomes varying as a function of the average safety value of each stimulus in the pair. After both stimuli were presented, subjects saw the outcome of the battle (*Safety Recognition*). Lost battles risked delivery of an electric shock to the subject's wrist (randomly delivered on 20% of lost trials). Successful battles resulted in 100% safety with no electric shock.

*Safety Value Updating* was tested in a separate, passive viewing task administered before and after the Safety Estimation Task. All stimuli from the Safety Estimation Task were presented in blocks of stimuli subsets depending on the safety value of each stimulus (e.g., the high safety block consisted of the weapons and animals with the highest safety probability). The first "naive" viewing was performed while subjects had no information about the Safety Estimation Task. The second "knowledgeable" viewing was performed after all runs of the Safety Estimation Task were completed and subjects had experienced the stimuli as relevant to their safety. Neural regions activated at the second viewing, but not the first viewing, were interpreted as tracking updated safety values. Dangerous weapons were used as protection during the Safety Estimation Task, meaning at the naive viewing all stimuli appeared dangerous, but, at the knowledgeable viewing, weapons with high safety value should be updated to indicate protection.

## Key concepts

Throughout we refer to several key concepts that are important to understand and interpret this work. The overarching investigation is to understand how humans estimate safety. **Safety Estimation** is the process by which humans evaluate the likelihood of safety (typically survival or absence of harm) in the presence of threats. Safety Estimation is central to understanding adaptive behaviors and defensive responses and occurs through the integration of information about both external threat information and internal protective capacities. **Safety Integration** is the process of merging distinct pieces of information (e.g., sensory stimuli, prior knowledge, or contextual factors) to create a coherent and actionable representation. This work refers to the theory that safety is not solely determined by the sensory features of a threat but also by self-relevant factors. **Meta-representation** is a process that occurs during integration whereby the brain constructs an internal model of 2 (or more) pieces of information, updating the representation of one piece of information that is not immediately perceptible based on new information received. Meta-representation occurs during cognitive integration, where higher-order neural regions such as the prefrontal cortex and association areas combine sensory

inputs with memories, predictions, and contextual cues. This concept aligns with several existing theories in cognitive neuroscience to explain how the brain achieves this complex task, including Bayesian Inference, Multisensory Integration, and Predictive Coding. **Safety Recognition** occurs when one can confirm safety where outcomes are known and **Safety Value Updating** occurs when one adjusts safety estimations based on new experiences or learning. Finally, we note that we refer to protective stimuli in this study as "**self-relevant**." This characterization is in *comparison* to the threat information, which is fully external to the subject. We acknowledge that the vmPFC is implicated in a myriad of self-related processes (e.g., linking episodic memories to form coherent self-concepts, moral reasoning, and social comparisons). For the purposes of the present work, we consider stimuli to be "self-relevant" when the value of a stimulus is reliant on how much a stimulus aids in fulfilling immediate, self-related goals, rather than on the stimulus itself.

## Results

Results are reported as facets of safety estimation, (1) *Safety Prediction* in response to the first stimulus presentation during the Safety Estimation Task when subjects could estimate safety based only on partial information; (2) *Safety Meta-representation* at the second stimulus presentation during the Safety Estimation Task when subjects could estimate safety on full information (Fig 1C); (3) *Safety Recognition* at the outcome during the Safety Estimation Task when subjects knew whether they were at risk of electric shock; and (4) *Safety Value Updating* comparing the "knowledgeable" and "naive" rounds of the passive viewing task. For each section, behavioral models are reported as well as univariate and multivariate fMRI results. Safety value (high versus low) and safety relevance (external versus self) are both considered.

### Safety prediction

Differences in *Safety Prediction* represented a bias toward initial safety information as a function of relevance (external versus self), given that safety probabilities for threat and protection were identical.

Behaviorally, subjects in both the behavioral and MRI samples tracked probabilities of winning and losing across the safety continuum such that subjects estimated a higher probability of winning for stimuli with higher safety probabilities (Fig 2A; Table A in S1 Text). Safety relevance (external versus self) affected initial safety bias such that subjects estimated greater safety when the first stimulus presented was protection, behavioral sample difference between protection and threat α = 0.23, 95% CI [0.21, 0.25]; MRI sample α = 0.28, 95% CI [0.25, 0.31] (Fig 2B).

Univariate fMRI analyses showed that the brain tracked safety value in response to the first stimulus presented, with dissociable circuits activated depending on safety value. As the stimuli increased in safety value, so did activation in the vmPFC, as hypothesized (Fig 4A and Table B in S1 Text). Considering each stimulus type separately, increasing safety for threatening animals activated the lateral occipital cortex (Fig 4B), and increasing safety for protection activated the vmPFC, amygdala, temporal pole, and anterior cingulate cortex (ACC) (Fig 4C).

Activation corresponding to decreases in safety was observed in the occipital pole and postcentral gyrus (all stimuli and threatening animals, **Fig A panels A and B in S1 Text**). Decreasing the safety value of protection did not show significant differences in parametric activation (**Fig A panel C in S1 Text**).

Informational Connectivity testing multi-voxel pattern synchronization between regions of interest (ROIs, Fig 6A) revealed a dense network involved in coding *Safety Prediction* (Fig 6B). The ACC (caudodorsal), insula (anterior), striatum (dorsal and ventral), vmPFC (anterior and posterior), thalamus, and hippocampus were connected during safety decoding. Betweenness

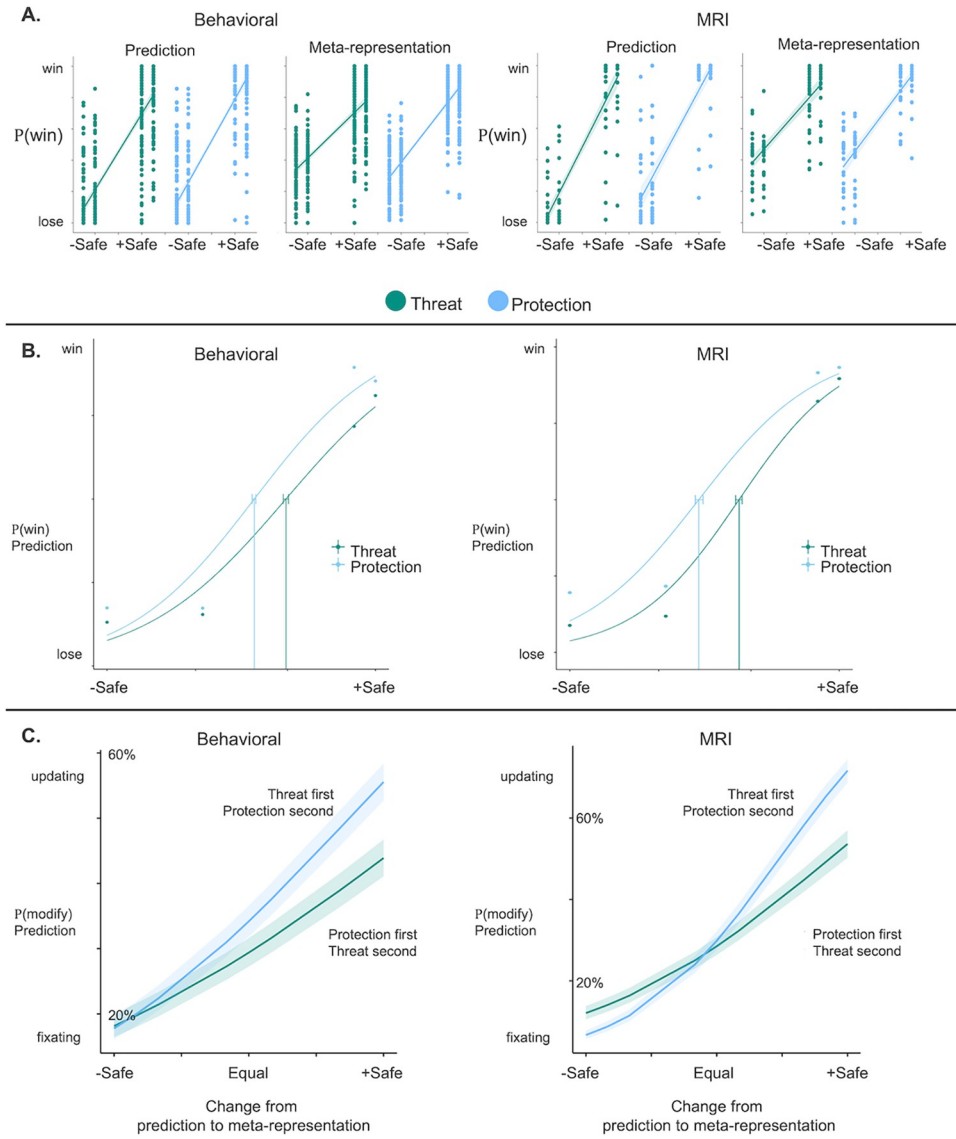

**Fig 2. Behavioral results showing safety estimation and biased predictions based on stimulus type.** (**A**) Mixed effects logistic regression depicting the association between experimentally established safety (x-axis) and subjective safety estimate derived from subject ratings for win/lose during battles (y-axis; 0 = lose, 1 = win). Threat and protection safety continuums are equivalent (x-axis). Results indicated subjects differentiated safety in accordance with the experimentally established safety continuum and tracked safety probabilities across the safety continuum for both stimulus types. See Table A in S1 Text. (**B**) Psychometric curves were fit for *Safety Prediction* in both samples. Subjects reached the safety detection threshold (α) faster when protection was presented as the first stimulus in the battle pair. Behavioral *N = 100* (left), MRI *N* = 30 (right). (**C**) Subjects were more likely to modify their safety estimations during *Safety Meta-representation* when threat stimuli were presented first followed by protective stimuli, especially when the safety value of the second stimulus changed to increase safety probability. Source data can be found at https://osf.io/8qg7y/ under "Behavioral data".

Centrality revealed the ACC as the primary hub of the network, with the insula, dorsal striatum, and posterior vmPFC also showing centrality contributions. While encoding the safety value of protective weapons, the anterior vmPFC emerged as the central hub, and the thalamus was eliminated as part of the network. While encoding safety in response to threatening animals, the posterior vmPFC emerged as the network hub and connections with the hippocampus, thalamus, and ventral striatum were no longer significant.

## Safety meta-representation

During *Safety Meta-representation*, stimuli were examined in terms of safety fluctuation: when the same stimulus had a higher-than-average safety value (e.g., a fist paired with a cat) compared with a lower-than-average safety value (e.g., a fist paired with a grizzly). Fluctuation in neural response reflects the integration of information about the initial stimulus encountered (e.g., cat, grizzly) with the second stimulus presented (e.g., fist) while holding the perceptual experience of the second stimulus constant.

Subjects' safety meta-representation behavior was consistent with their predictive behavior such that subjects in both samples estimated a higher probability of winning for stimuli with higher safety probabilities (Fig 2A and Table A in S1 Text). There was no significant difference in the threshold of safety detection as a function of the second stimulus type, behavioral sample difference between protection and threat α = 0.01, 95% CI [−0.03, 0.04]; MRI sample α = −0.02, 95% CI [−0.09, 0.04].

During *Safety Meta-representation*, subjects fixated more on the initial safety value if protection stimuli were presented first. In other words, subjects were more likely to switch their safety prediction from win to lose or lose to win when threats were presented first and protection second (stimulus type: behavioral B = 0.23, *SE* = 0.02, *z* = 11.20, *p* < 0.001; MRI B = 0.15, *SE* = 0.03, *z* = 4.04, *p* < 0.001), particularly when the protection information increased the likelihood of safety (interaction between stimulus type and safety change: behavioral sample B = 0.04, *SE* = 0.006, *z* = 6.84, *p* < 0.001; MRI sample B = 0.12, *SE* = 0.01, *z* = 9.59, *p* < 0.001, Fig 2C). Overall, subjects rated winning probabilities as higher if the second stimulus was a powerful weapon following a dangerous animal, compared to a weak weapon following a safe animal (75% versus 53%), despite equivalent experimentally established safety probabilities (Fig 3A). However, subjects rated safety probabilities as equivalent when a dangerous animal followed a weak weapon or a safe animal followed a powerful weapon (63% for both) (Fig 3B).

When stimuli increased in safety value compared to its average, vmPFC activation increased parametrically (Fig 4D and Table B in S1 Text), evincing a more distributed pattern of activation across the vmPFC compared with *Safety Prediction*. Considering each stimulus type separately, neural response to external threats increased in safety (as a function of being paired with more powerful weapons) activation increased in the vmPFC and lateral occipital cortex (Fig 4E). As self-relevant protective stimuli increased safety, neural response increased in the occipital cortex (Fig 4F). Conjunction analyses identified the vmPFC as a common neural substrate of *Safety Prediction* and *Meta-representation* (Fig 4G).

In response to increasing danger, activation increased in the insula, thalamus, ACC, and PAG (**Fig A panel D in S1 Text**), showing a more typical pattern of defensive circuitry activation than that observed in response to dangerous stimuli at first presentation. As threatening animals become less safe, the bilateral thalamus, right insula, pre-supplementary motor cortex, and medial occipital cortex parametrically responded with increased activation (**Fig A panel E in S1 Text**). When self-protective stimuli were rendered less safe than average because of the initial danger value of the threat, neural response increased in the insula, thalamus, and ACC (**Fig A panel F in S1 Text**).

Testing multi-voxel pattern synchronization using Informational Connectivity, the same safety network was decoded with the addition of the amygdala in response to *Safety Meta-representation* (Fig 6C). Informational Connectivity during *Safety Meta-representation* tracked changes in safety from overall average values, in line with univariate results. Betweenness Centrality revealed a switch from the ACC as the top *Safety Prediction* hub to the dorsal striatum as the top *Safety Meta-representation* hub when examining all stimuli and in response to threatening animals, with the ACC, insula, and posterior vmPFC playing centrality roles. The

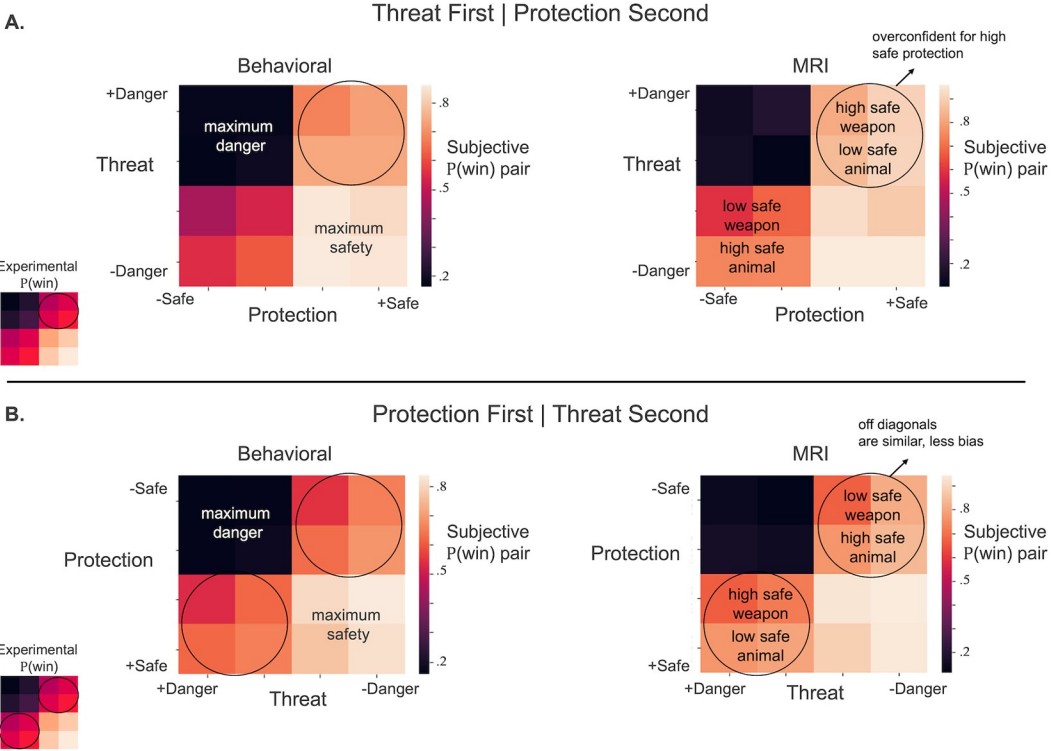

**Fig 3. Behavioral results showing safety estimation updating as a function of stimulus presentation order.** (**A**) When threat was presented first and protection second, subjects were overconfident about their safety estimates when armed with a high-value weapon. (**B**) When protection was presented first and threat second, subjects were more likely to differentiate according to the threat pairing in line with experimental probabilities. Heatmap scales for subjective safety ratings are normalized for responses in each of the Behavioral and MRI samples. Source data can be found at https://osf.io/8qg7y/ under "Behavioral data".

posterior vmPFC was the central hub for the *Meta-representation* of protective weapons, followed by the dorsal striatum and ACC and the insula no longer showing as a central hub.

## Safety recognition

*Safety Recognition* was tested in response to the trial outcome when subjects won the battle and were safe from electric shock. Activation in the vmPFC as well as in the striatum and bilateral hippocampus increased in response to safe outcomes during the task (Fig 4H).

In response to negative outcomes (lost battles conferring risk of shock), the bilateral insula and PAG were activated (**Fig A panel G in S1 Text**).

## Safety value updating

*Safety Value Updating* was examined during a separate fMRI task. Subjects viewed all threat and protection stimuli in a rapid block design before the Safety Estimation Task (naive first viewing) and then again after performing the full Safety Estimation Task (knowledgeable second viewing). This allowed us to test whether subjects updated their response to stimuli with high safety values after experiencing the stimuli during the Safety Estimation Task. After *Safety Value Updating*, targeted multivariate searchlight revealed significant changes in vmPFC representation of stimuli with high safety probabilities (cat, goose, gun grenade) (Fig 4I). No significant changes from pre- to post-task emerged for stimuli with high danger probabilities (lion, grizzly, fist, stick).

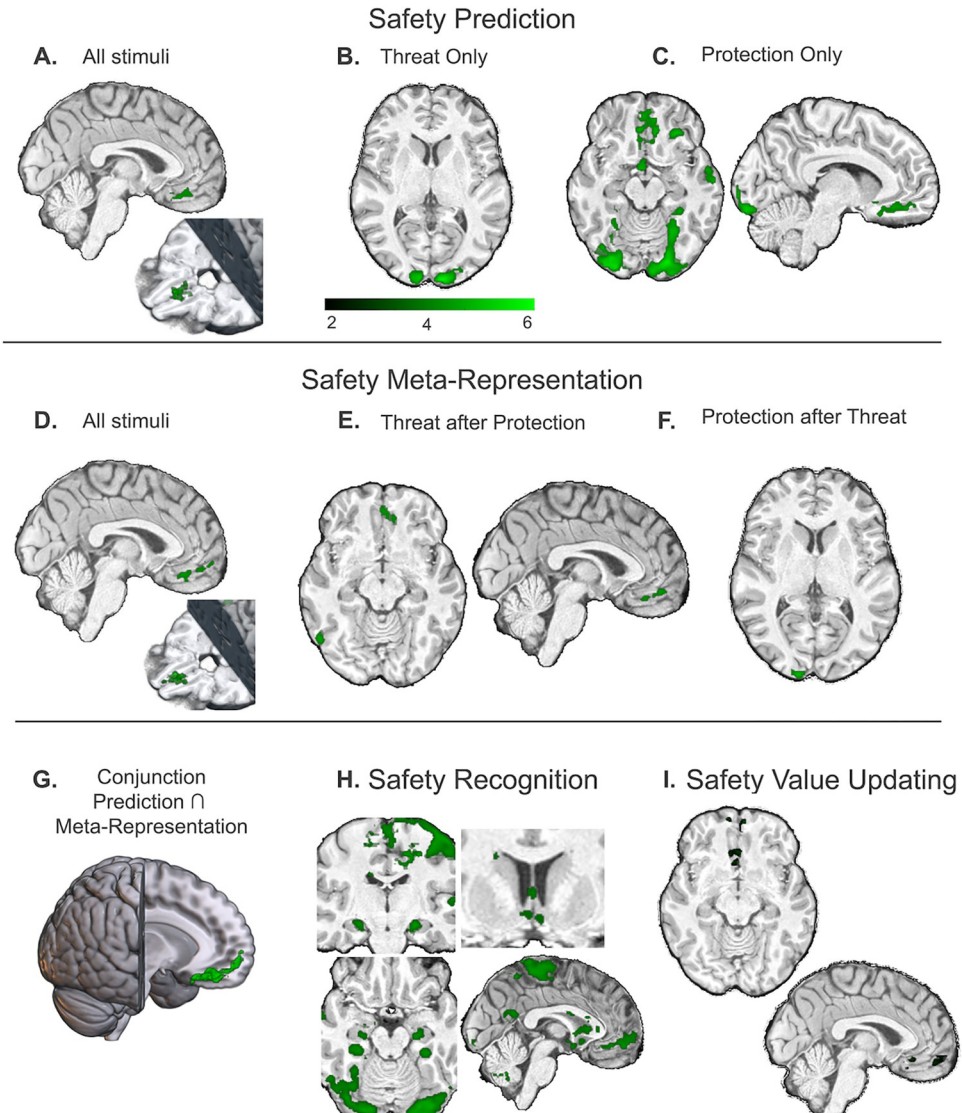

**Fig 4. Neural response to safety increases during each task phase, highlighting the role of the vmPFC in responding to safety increases and protection stimuli.** Analyses in were conducted using FSL Randomise, TFCE, FWE-corrected $p < 0.05$. Color bar indicates t-intensity values. (**A–C**) Parametric increases in whole-brain neural activity that track increases in experimentally established safety value of stimuli during **Safety Prediction**. The first stimulus presented represented a bias to partial information, which measures a differentiation in neural activity as a function of stimulus type (threat versus protection). Significant clusters indicate activation increased in those regions as safety probability increased. Safety increase was based on the average experimentally established safety probability of each stimulus (protection continuum order: fist, stick, gun grenade; threat continuum order: cat, goose, lion, grizzly). (A) Threat and Protection collapsed, (B) Threat only, (C) Protection only. (D–F) Parametric increases in whole-brain neural activity that track increases in experimentally established safety value of stimuli during Safety Meta-representation. The second stimulus safety value was based on the combined safety probability of the first and second stimuli. For analyses, safety was based on comparison with the average safety value of the stimulus. For example, if a stick was shown as the second stimulus and was paired with a cat, the probability of safety would increase from 35.72% (safety average for all stick trials) to 57.14% (safety when stick is paired with cat) (see Fig 1B). (D) Threat and Protection collapsed, (E) Threat only, (F) Protection only. (G) Conjunction of Shared Activation between Safety Prediction and Safety Meta-representation. Conjunction analyses for the safety prediction phase of increasing safety versus increasing danger and shared activation with the safety meta-representation phase of increasing safety versus increasing danger. All stimuli represented. Results indicate overlapping activation in the vmPFC; $Z = 2.3$, $p < 0.05$. (H) Neural activation in response to Safety Recognition. Analyses focused on the outcome screen after subjects learned they had been successful during the battle and had achieved 100% certainty of safety compared with unsuccessful battles when the outcome screen indicated potential for electric shock. (I) Safety Value Updating. Multivariate

searchlight revealed neural activation change in the vmPFC when subjects engaged in Safety Value Updating for the high safety block (see Fig 1D). Analyses examined the contrast of the Knowledgeable version (post-task viewing) versus the Naive version (pre-task viewing) (E). Searchlight was a priori restricted to the vmPFC using an ROI defined via Neurovault. Source data can be found at https://osf.io/8qg7y/ under "MRI data." ROI, region of interest; vmPFC, ventromedial prefrontal cortex.

### vmPFC

Results show vmPFC involvement at all stages of safety estimation (Fig 5) with posterior anterior gradients as safety increases in certainty from *Prediction* (partial information) to *Meta-representation* (full information, outcome unknown) to *Value Updating* (no shock possible, stimulus value learned) to *Recognition* (full certainty of safety). Safety activation in the vmPFC is primarily identified in area 14m with safety updating and recognition (conditions during which safety was guaranteed) extending to area 10 (frontal pole) [34].

## Discussion

This study identifies neural systems involved in safety coding, provides evidence that *Safety Prediction* evokes dissociable circuits depending on whether the stimulus has self-relevance, and supports the hypothesis that the brain integrates threat and protective information to *Meta-represent* safety. The vmPFC emerged as a robust hub of human safety coding during safety estimation, including *Safety Prediction*, *Meta-representation*, *Recognition*, and *Value Updating*. The vmPFC showed specific tuning to protective information, supporting the importance of developing models of safety computing to expand beyond extinction of external threat.

During *Safety Prediction*, subjects were quicker to detect safety when presented with self-relevant protective stimuli compared to when presented with externally relevant threat stimuli. Neurally, vmPFC activation parametrically increased as protection increased in safety value. Threat stimuli, in contrast, activated sensory and defensive neural systems. Despite equivalent experimentally established safety probabilities for threat and protection stimuli, only protection evoked activation in the vmPFC. One intriguing possibility for vmPFC sensitivity to protective weapons, beyond their safety relevance to the self, is that the vmPFC was coding

## vmPFC Activation for All Task States

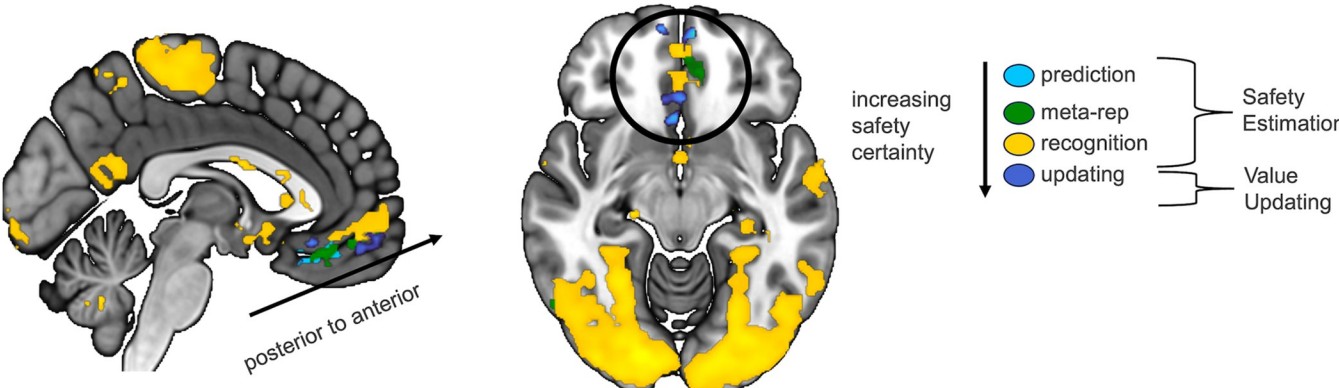

**Fig 5. vmPFC overlap for all stages of Safety Estimation.** Results indicate a posterior to anterior shift as safety becomes more certain. Source data can be found at https://osf.io/8qg7y/ under "MRI data." vmPFC, ventromedial prefrontal cortex.

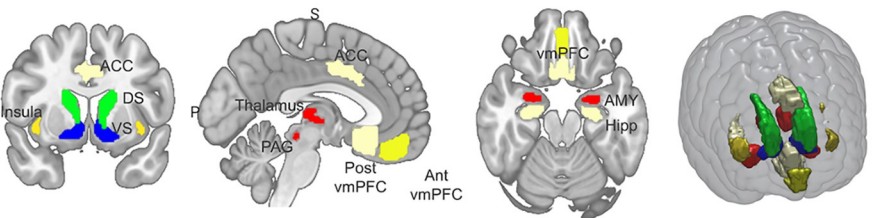

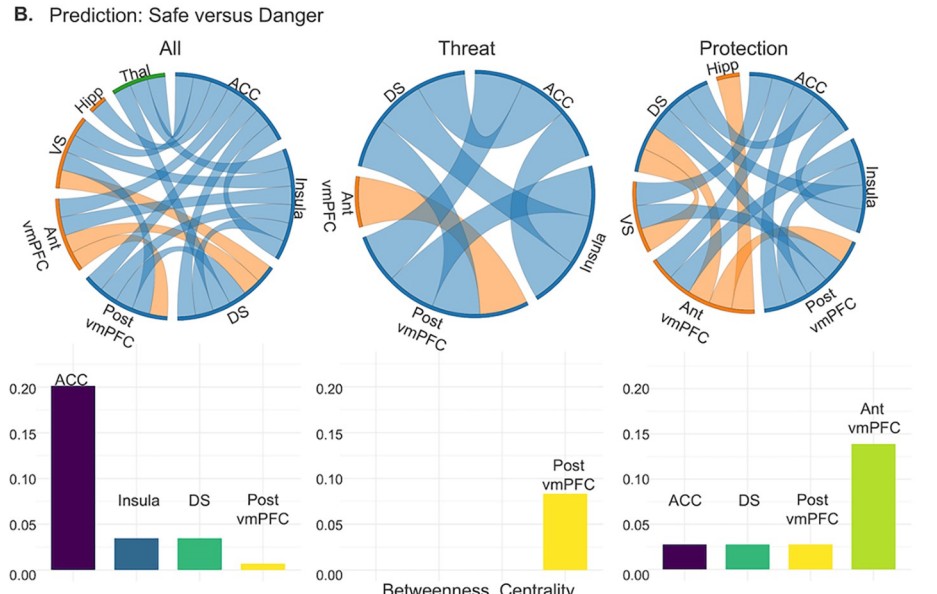

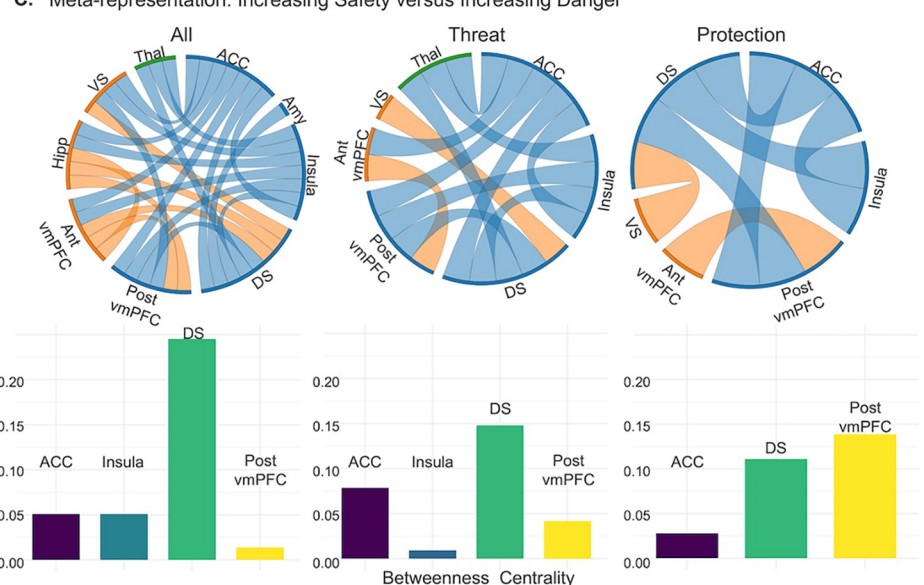

**Fig 6. Multivariate functional network connectivity for trials of increased safety value versus those with increased danger value among a network of 10 a priori selected ROIs identified as relevant for threat/safety identification in the existing literature.** (**A**) ROIs used in analyses. Multivariate functional connectivity (Informational Connectivity) was computed by using covariation trial-by-trial decoding accuracy between each pair of regions. (**B**) Informational Connectivity analyses resulted in a connectivity matrix between ROIs for the first stimulus presentation indicating regions that communicated while decoding states of safety for all stimuli collapsed, threat stimuli, and protection

stimuli (left to right), all connections $p < 0.05$ corrected. We computed betweenness centrality within this network to find hubs connecting regions during decoding. (**C**) For second stimulus presentation, regions of connectivity were decoded based on whether stimuli increased in safety or increased in danger as a function of the stimulus pairing, all connections $p < 0.05$ corrected. For example, a lion paired with a fist would increase in danger, whereas a lion paired with a grenade would decrease in danger. Source data can be found at https://osf.io/8qg7y/ under "MRI data." ACC, anterior cingulate cortex; ROI, region of interest; vmPFC, ventromedial prefrontal cortex.

ownership associations between objects and the self [31]. We did not directly test whether ownership was part of the cognitive milieu in our study, but our findings are consistent with the perspective that the vmPFC assigns personal significance to objects based on its meaning and function for the self [35]. Our findings agree with prior literature establishing that the vmPFC tracks self-related associations beyond reward-based learning [35] and point to a specialized role of the vmPFC in tracking self-relevant information during safety estimation.

As threat stimuli increased in danger during *Safety Prediction*, so did activation in the occipital pole and postcentral gyrus. We interpret this activation to reflect several processes related to visual, attention, and sensory processing when evaluating stimuli with heightened salience, like those representing a threat. The occipital pole is involved when visual processing demands are enhanced, as is the case when evaluating threatening stimuli that capture attention more effectively—as is shown in prior work suggesting threatening stimuli are prioritized in the visual system [36,37]. The postcentral gyrus contains the primary somatosensory cortex and is involved in processing somatic sensory information, which may reflect an embodied response to perceived threat preparing for potential defensive action [38]. Enhanced sensory processing in both visual and somatosensory areas may thus serve as an adaptive mechanism to improve threat detection and prepare for action.

During *Safety Meta-representation*, subjects were again quicker to detect safety value for protection. Neurally, subjects meta-represented the first stimulus when evaluating the second stimulus, despite the absence of perceptual information about the first stimulus [34,39]. In response to threat stimuli, the same vmPFC region that activated to protection during *Safety Prediction* was activated. In response to protection stimuli, the same sensory regions of the visual cortex that activated to threat during *Safety Prediction* were activated. In other words, the pattern of activation at the second stimulus was inverted, which we interpret as meta-representation during safety integration. This process of reactivation/replay has been observed in aversive learning in humans [34,40]. We observed patterns of activation at a whole-brain level, demonstrating striking similarities to the initial state encountered. We did not conceptualize meta-representation as replay in our study because our task does not afford the typical planning opportunity that is tested during traditional examinations of replay. Our interpretation is bolstered by the task design: stimuli at the second stimulus pairing were perceptually identical and only differed in safety value as a function of their pair. Thus, the neural systems responding to increases in safety during *Meta-representation* were responding to changes in the safety value as a function of the first stimulus safety value and its resulting influence on the overall safety probability. Similar behavioral patterns emerged: Subjects were more likely to update predictions during *Safety Meta-representation* when self-relevant protection conflicted with initial external threat information. On the other hand, when protection was presented first, subjects did not shift safety estimations. Our results fit with a broader body of literature showing that memory retrieval induces aspects of the pattern of neural activity evoked by the original stimulus presentation [39,41].

As threat stimuli increased in danger during *Safety Meta-representation*, so did activation in defensive circuitry, including the insula, thalamus, ACC, and PAG. Parametric increases in activation of the thalamus, insula, pre-supplementary motor cortex, and medial occipital

cortex suggest heightened sensory and motor preparation to manage escalating danger. This aligns with models of threat appraisal, where sensory processing regions and motor planning systems engage to anticipate defensive actions. These responses reflect a dynamic and context-sensitive process, consistent with the brain's reliance on interconnected defensive circuits to adapt to changes in safety.

Our findings converge with prior work indicating the vmPFC is involved in updating mental representations of threat and reward particularly in contexts like fear extinction [17]. Given vmPFC involvement in the physiological regulation of autonomic processes like heart rate and skin conductance [42,43], it is likely that vmPFC engagement we saw during our task reflects a more complicated myriad of processes that we did not explicitly measure, including emotion regulation [44,45]. Additionally, our behavioral results show safety recognition varied as a function of stimulus type, and the vmPFC was preferentially activated to protection during the Safety Estimation Task. These findings are consistent with the vmPFC playing a role in self-relevant processes specifically rather than undifferentiated emotion regulation broadly. If the vmPFC were primarily tracking the latter, we would expect to see consistent activation regardless of stimulus type because both stimuli presentations occur during a need for regulatory engagement. Our findings fit with an interpretation of the vmPFC as providing a substrate for linking experiences, contexts, and events to the bio-regulatory or emotional state of the self [32,46].

Given the lack of information that classic univariate connectivity approaches have (i.e., PPI), we used informational connectivity to examine synchronization of voxels. Multivariate connectivity revealed a safety network consisting of the anterior and posterior vmPFC, dorsal and ventral striatum, caudodorsal ACC, and insula. The hippocampus, thalamus, and amygdala also emerged as connected to this core network, but with less consistency depending on stimulus relevance. In response to *Safety Prediction*, threat and protection networks showed a shift in hub organization from the posterior to anterior vmPFC. The dorsal striatum emerged as a core hub for *Safety Meta-representation*. The dorsal striatum has been previously linked to punishment-based avoidance, with dorsal striatum damage resulting in suboptimal defensive choice [47]. We situate our findings with consideration of this prior work and interpret *Safety Meta-representation* as necessary to generate choices about defensive action. The ventral striatum and hippocampus were both connected to the safety network during protection evaluation but not during threat evaluation. The vmPFC's coupling with salience (insula) and reward regions (striatum) links self-relevant safety processing with affective and reward systems, suggesting its role in mediating the personal significance of stimuli and maintaining a sense of self across contexts [48]. This is also consistent with our prior work demonstrating that safety conferred through protection is distinct from threat despite both occurring in aversive contexts [49], likely due to its self-referential signaling. The PAG did not emerge as part of the safety network, consistent with its role in fast innate defensive reactions [13,50]. It did however emerge preceding risk of shock both during outcomes for lost battles and under increasing danger during *Meta-representation*.

Battle outcomes were tested as *Safety Recognition* and served as the purest test of safety neural circuitry. Neural activation in response to safety certainty, when subjects won the battle and were 100% safe from shock compared to when they lost battles, increased in the vmPFC, striatum, and hippocampus. In response to lost battles when there was risk of electric shock, compared with won battles, subjects demonstrated increased engagement of canonical defensive circuitry in the PAG and bilateral insula. Activation of the striatum and hippocampus indicates subjects learned during outcomes but did not reengage these circuits during prediction. The vmPFC, however, was engaged in response to both *Safety Prediction and Recognition*. We interpret the difference in vmPFC engagement compared with striatum and hippocampus

engagement to indicate a more general role of the vmPFC in recognizing and predicting safety, rather than tracking outcomes to reinforce learning. This interpretation is further supported by using stimuli that had known and biologically relevant danger values (predators and weapons), which likely attenuates online associative learning during the task because the danger of the stimulus is already coded as part of its inherent properties [51]. The brain should treat these stimuli as known [52] and, therefore, vmPFC activation is unlikely a result of generating new learning effects in response to incidental shock pairings.

The vmPFC was identified through searchlight analyses as involved in *Value Updating* for high-safety stimuli. The vmPFC was differentially engaged after experience during the Safety Estimation Task compared to naive viewing. Importantly, the stimuli in this study were all perceptually threatening and therefore could all have been interpreted as dangerous when subjects viewed them without knowledge of the Safety Estimation Task. Dangerous weapons have a general threat connotation and only take on a safety status when wielded to protect oneself. All animals presented as threats were shown to be attacking and angry (as opposed to a cuddly housecat). Thus, we expected that all stimuli would be represented as threatening before exposure to the stimuli as relevant for personal safety during the Safety Estimation Task. Changes in multivariate vmPFC representation after experience with the stimuli provide converging evidence that the vmPFC integrates information about safety rather than processing a more general stimulus value.

We identified a safety coding network that included subcortical and cortical regions involved in diverse processes including learning, reward valuation, and affect signaling. Although this study makes a significant advance in understanding how the brain contributes to estimating safety, many questions remain. First, how universal is the role of the vmPFC in coding safety? This study used a model of predator–prey interactions. Although humans are not typically exposed to predation, everyday threats induce neurobiological and psychophysiological states like those observed under predatory threats [3,13,53]. Using outwardly dangerous animals as threats eliminated interference from prior social experiences and allowed us to test the neural systems involved in *Safety Value Updating*. However, future work should examine safety coding during a diversity of threats including complex human interactions. Second, how do safety network communications evolve as a function of spatiotemporal dynamics? Our prior work shows that neural systems involved in defensive responding are dissociable along the threat imminence continuum [50,54]. The task used in the current study was not designed to examine dynamic threat nor did it evoke escape behavior. Further work is needed to determine under what conditions vmPFC-supported cognition is unavailable. Third, how do affect dynamics influence safety estimation? We did not collect participant report on their affective response to stimuli during the task. As a result, we cannot be certain whether our findings reflect an affective or non-emotional component of safety processing. Other work indicates a role of the vmPFC in the extinction of emotional arousal [55], and future work should consider how that role generalizes to safety estimation. Lastly, our MRI sample size prevented the examination of brain-based individual differences in age and psychopathology. Adolescence is a critical time to study safety computations given the prevalence of anxiety disorders, changes in metacognitive abilities, poorer threat-safety discrimination compared with adults, and imbalance in amygdala–vmPFC contributions to safety processing [24,56–59]. These features of adolescent development may result in impaired self-relevant safety processing.

The vmPFC coded safety during all task states from *Safety Prediction* to *Meta-representation* to *Safety Recognition* and *Value Updating*. Activation to these states was primarily identified in area 14m, with Safety Recognition and Value Updating extending to area 10. We identified what appeared to be a gradient from the posterior to the anterior part of vmPFC area 14m with activation extending more anterior as safety increased in certainty. This apparent gradient

supports the possibility of subparcellation of area 14m into smaller areas with posterior and anterior distinction. During detailed mapping of vmPFC architecture, Mackey and Petrides note that "the posterior part of area 14m is less well developed and is less granular than the anterior part" [34]. This aligns with our recent proposition [2] that the posterior vmPFC may encode simpler representations of threat, whereas the anterior vmPFC may encode more complex safety associations. This apparent gradient aligns with the broader organizational gradient of the prefrontal cortex, where anterior regions are generally more granular and complex, reflecting their role in higher-order cognitive and integrative functions. Our findings support assertions that the anterior vmPFC plays a role in integrating the value of self-relevant stimuli to influence the higher-order construction of affective processes, including safety [2,60].

Beyond identifying how the human brain codes safety, our findings have potential implications for improving clinical interventions for anxiety. Current therapies focus on threat extinction but are ineffective for up to 50% of individuals [61]. A major problem with studying safety through the lens of threat extinction is the assumption that safety is the inverse of threat (in the absence of the aversive event the stimulus itself becomes "safe"). This confounding association does not consider how safety fluctuates independent of threat, for example, when protective resources can change safety while external threats remain unchanged. Current therapies target extinguishing fear responses to threats [62], but our data suggest focusing on self-relevant safety cues may be a promising therapeutic avenue. Also supporting a departure from extinction-focused approaches, recent work showed repetitive transcranial magnetic stimulation (rTMS) modulation of the anterior mPFC inhibited implicit fear reactions to learned threats [63]. This is a departure from emphasis on the dorsolateral PFC as a regulatory hub, which may be limited to extinction paradigms [64,65]. Intriguingly, the mPFC is a hub of the brain's default mode network (DMN), which point to safety as an aspect of baseline human cognition. Psychopathologies, like anxiety, are often characterized by DMN dysfunction [66,67], which may mechanistically explain co-occurring deficits in safety estimation. Our findings provide a neuroscientifically grounded framework of safety beyond threat extinction and set the stage for future research to better understand how the human brain adaptively codes safety.

Finally, our work is situated in a broader understanding that the vmPFC supports self-referential processing, extending this role to safety estimation. The brain is engaged in considerable focus on tracking associations related to the self [31,32,35,46,48,68]. Our study demonstrates this does not just occur in conspecific social interactions or in identifying which objects belong to oneself, but also with respect to evolutionarily conserved threat appraisal. When the self is under threat, the vmPFC may help integrate threat-related stimuli into the broader narrative of self-representation, shaping responses based on past experiences and perceived safety. This raises a compelling case for considering how altered vmPFC activity in affective clinical disorders reflect disruptions in assessing and assigning personal significance to events, potentially tied to deficits in views of the self's capacity to establish safety.

## Methods

### Behavioral

One hundred thirteen human subjects completed the Safety Estimation Task online, which was identical to the task performed in the MRI scanner except that it was performed in a single continuous session for the Behavioral subjects. The Behavioral subjects did not complete the Safety Value Updating Task because there were no decision components for this task, so no meaningful data could be collected. Behavioral subjects were recruited through Prolific, a recruitment and data collection platform that produces high-quality data [69]. Seven subjects

responded to fewer than 20% of trials and 6 subjects made safety choices that were inversely related to the safety continuum (i.e., judging safe stimuli as dangerous and dangerous stimuli as safe) resulting in an accuracy of more than 3 standard deviations (SDs) below the group mean. Excluding these 13 subjects resulted in a final behavioral sample of 100 subjects ($M$age = 29.20 years, $SD$ = 6.61, range = 19–40, 50 females 51%).

## MRI

Thirty-one human subjects completed the same Safety Estimation Task as the Behavioral sample. The only difference between the Behavioral and MRI samples is that the MRI sample completed the Safety Estimation Task in 4 blocks over 2 days while undergoing functional MRI. Separating the MRI sessions into 2 days was designed to alleviate motion problems from subjects spending too long in the scanner without a break. MRI subjects also completed the Safety Value Updating Task described below. MRI subjects were recruited through flyers and advertisements. One subject had +3SD below the group mean in accuracy and was excluded, resulting in a final MRI sample of 30 subjects ($M$age = 27.83, $SD$ = 4.86, range 20 to 40 years, 15 females 50%). One additional subject was excluded from analyses relating to the passive viewing task (*Safety Value Updating*) due to poor registration and dropout in the vmPFC, the primary area of interest. The full study was conducted for approximately 90 min per day over 2 days.

## Inclusion and exclusion criteria

Inclusion criteria for both samples were age 18 to 40, fluent in English, and normal or corrected vision. The MRI sample was additionally required to have no psychiatric or neurological illness and be eligible for MRI, including having no metal contraindications.

## Ethics

All methodology was approved by the California Institute of Technology Internal Review Board (protocol 21–1127) and was conducted according to the principles expressed in the Declaration of Helsinki. All subjects consented to participation through written consent. Subjects were compensated for their time.

## Procedure

For MRI sessions, subjects first provided informed consent. Outside of the scanner, physiological equipment was attached and a shock workup procedure was conducted. Electrodes were placed to the underside of the wrist 1 to 2 inches below the palm. During the shock workup procedure, shocks started at a low intensity and increased to the level the participant considered "uncomfortable but not painful" using a 0 to 10 discomfort scale (0 = "not at all," 5 = "moderately," and 10 = "very," $M_{session1}$ = 4.87, $SD_{session1}$ = 0.34; $M_{session2}$ = 5.16, $SD_{session2}$ = 0.56). Shock intensity from session 1 was highly correlated with shock intensity from session 2 at $r(31)$ = 0.93. Shocks were delivered using STMISOC with 2 LEAD110A (BIOPAC, Inc.) and 2 Telectrode T716 Ag/AgCl electrodes. The shock consisted of 2 pulses 0.03 s apart delivered during outcome screens for lost battles.

While in the scanner, subjects first completed the passive viewing task to maintain ignorance to stimuli relevance. Next, subjects completed instructions for the Safety Estimation Task and 10 practice trials. During the first session, subjects completed a structural MPRAGE and 2 runs of the Safety Estimation Task. During the second session, subjects completed 2

runs of the Safety Estimation Task. After all Safety Estimation Task runs, subjects completed the passive viewing task again.

## Safety estimation task

During the Safety Estimation Task, subjects played a series of battles in which they attempted to defeat an animal with a weapon. Subjects did not receive a choice in animal or weapon, and probabilities of winning or losing the battle were experimentally established (Fig 1A and 1B). We experimentally established safety probabilities to control comparisons between types of stimuli (threat versus protection) along the safety continuum, but we acknowledge that these probabilities are not definitive or universally applicable outside of the experimental setting. Stimuli were recognizable to reduce learning confounds (e.g., it is widely known that a grizzly is more dangerous than a goose). For each trial, subjects saw 2 images presented for up to 6 s maximum. Image presentation offset with subject response. Each image was separated by an interstimulus interval (ISI) presented for 2 to 5 s with duration jittered. Each pair of images contained 1 weapon and 1 animal, with the presentation order counterbalanced. Subjects were told to indicate with a button press while the stimulus was on screen whether they thought they would win or lose the battle against the animal with the weapon provided. Subjects used their right index finger to indicate a win prediction and their right middle finger to indicate a loss prediction. As soon as predictions were made the stimulus offset and the ISI was presented. Subjects were told "winning does not necessarily mean killing the other animal. You can interpret winning as defeating the other animal either because it retreats or because it is physically defeated." Response to the first image presentation was based on partial information, whereas response to the second image presentation was based on full information of the animal/weapon pair. Animals and weapons ranged in safety value on a 4-point continuum with matched contingencies (Fig 1B). Likelihood of win/loss depended on the combined probability of the animal/weapon. For example, if subjects encountered a lion, they had an average 64.29% likelihood of losing the battle regardless of the weapon. That likelihood increased to 78.57% if subjects were equipped with a stick and reduced to 42.86% if subjects were equipped with a grenade. After both images were presented, subjects saw the outcome of the battle for 2 s. For the MRI sample, subjects had a 20% chance of receiving an electric shock to the wrist for every lost battle. Subjects were 100% safe from electric shock if they won the battle. For the behavioral sample, points were lost and gained depending on battle outcomes. Trials were separated by an inter-trial interval (ITI), presented for 1 to 6 s with duration jittered. All subjects completed 448 trials. The behavioral sample completed trials in a single session, whereas the MRI sample completed 4 runs of 112 trials each run each over 2 days. The first 2 runs were presented on day 1, with an average of 26 min from the start of Run 1 to the start of Run 2. The second 2 runs were presented on day 2, which occurred on average 1 day and 7 h after day 1, and the time between the start of Run 3 and the start of Run 4 was 25 min on average. The Safety Estimation Task was programmed using PsychoPy v2021.2.3.

## Stimuli development

Prior to data collection, a series of stimulus development tests were conducted. Sixty subjects participated in stimuli development. Data from 2 subjects were excluded due to failure of attention checks resulting in a development sample of 58 ($M$ age = 23.07 years, $SD$ = 4.50, range = 18–38, 39 females 67%). Twenty animals and 20 weapons were presented in paired head-to-head battles. For animal head-to-heads, subjects were asked to pick which animal they thought would win in a battle. Subjects were given the same instructions as the Safety Estimation Task: "winning does not necessarily mean killing the other animal. You can interpret

winning as defeating the other animal either because it retreats or because it is physically defeated." For weapon head-to-heads, subjects were asked to pick which object they thought was more powerful. Subjects were told, "You can think of this as choosing which weapon would win in a head-to-head battle." The danger of each animal was also rated on a 0 to 100 scale (0 = not at all dangerous, 100 = extremely dangerous), and the power of each weapon was rated 0 to 100 (0 = not at all powerful, 100 = extremely powerful). Based on the results of these inquiries, a final set of 4 animals and 4 weapons were selected with 2 of each at the high end of the safety continuum and 2 of each at the low end. Stimuli were reduced to 4 based on scan time considerations and the number of trials needed for multivariate analyses. Lion and grizzly were rated as the most dangerous stimuli and cat and goose were rated as the second and third least dangerous stimuli (rat was selected as the least dangerous but ultimately excluded from the set to avoid conflating threat with disgust) (**Fig B panels A and B in S1 Text**). The same rankings were reported for the head-to-head battles across all animals. The grenade and gun were rated as the most powerful weapons and as most likely to win head-to-head (**Fig B panels C and D in S1 Text**). Fist and stick were rated in the bottom 30% of power ratings and bottom 20% of head-to-heads. Other weapons rated as less powerful were excluded due to concerns of unwieldy usage (i.e., rope) (**Fig B panels C and D in S1 Text**). Additional information on selection of animals and weapons is provided in **Fig B in S1 Text**.

## Safety value updating

Before and after the Safety Estimation Task, the MRI sample completed a passive viewing task. During the task, subjects saw animal and weapon images that were used in the Safety Estimation Task in a series of blocks. Subjects were instructed to look at each image carefully and that no decisions were required and no shocks would be administered. Images were presented for 0.5 s per image. Images were presented 5× in each block. Six blocks were repeated 5× each. Each stimulus image was followed by a 3 s ISI (interstimulus interval breaking up the presentation of individual stimuli) and each 20-trial set within a block was followed by a 12-s ITI (intertrial interval breaking up entire blocks). This approach was selected as it has been shown to be a validated approach for the localization of neural regions specifically responsive to stimuli recognition [70]. Blocks were comprised of high danger stimuli with high shock probability during the Safety Estimation Task, high safety stimuli with low shock probability during the Safety Estimation Task, high threat stimuli with high external threat value outside of the experimental environment, low threat stimuli that have low external threat value outside of the environmental experiment, low threat stimuli with low external threat value, weapons, and animals.

## Behavioral models

Psychometric curves were fit to examine safety prediction as a function of stimulus type (protection, threat) (Fig 2B); α parameters represent the threshold level where safety estimation reached a 50% probability. General effect sizes are reported as 95% confidence intervals. Binary logistic mixed effects models were fit to examine whether (1) subjects tracked the experimentally established safety continuum with subjective estimations of winning and losing during safety prediction and safety meta-representation (Fig 2A); (2) subjects modified safety estimation during safety meta-representation as a function of stimulus presentation order (protection first versus threat first) (not depicted graphically); and (3) safety change (increasing or decreasing from average) moderated the effect of stimulus presentation order on safety estimation modification (Model 2) (Fig 2C). Mixed effects models were estimated using R (version 4.1.3) and the lme4 package (version 1.1.28) [71,72].

## MRI data acquisition

Functional and structural data were acquired using a Siemens 3 Tesla Magnetom Prisma MRI scanner fit with a 32-channel head coil. For the acquisition of the functional images, we used a T2* weighted gradient EPI sequence. The repetition time (TR) was 1.12 s, the echo time (TE) was 30 milliseconds, the flip angle was 54 degrees, and the voxel resolution was $2 \times 2 \times 2$ mm. A total of 512 slices were acquired in ascending interleaved order with a multiband acceleration factor of 4. Each functional run consisted of 1,279 volumes. For the structural data, we used a T1*-weighted MPRAGE sequence (image size $208 \times 256 \times 256$ voxels, TR 2.55 s, TE 0.16 ms, flip angle 8, slice thickness = 0.9 mm).

Stimuli were projected onto a flat screen mounted in the scanner bore. Participants viewed the screen using a mirror mounted on a 32-channel head coil. Extensive head padding was used to minimize participant head motion and to enhance comfort. Participants made their safety judgments with their right hand using a 4-finger-button response box.

## MRI preprocessing

Raw data were converted from DICOM to BIDS format. Results included in this manuscript come from preprocessing performed using fMRIPrep 21.0.0 [72,73] (@fmriprep1; @fmriprep2; RRID:SCR_016216), which is based on Nipype 1.6.1 (@nipype1; @nipype2; RRID: SCR_002502). A B0-nonuniformity map (or fieldmap) was estimated based on 2 (or more) echo-planar imaging (EPI) references with "topup" (@topup; FSL 6.0.5.1:57b01774). The T1-weighted (T1w) image was corrected for intensity non-uniformity (INU) with "N4Bias-FieldCorrection" [@n4], distributed with ANTs 2.3.3 [@ants, RRID:SCR_004757], and used as T1w-reference throughout the workflow. The T1w-reference was then skull-stripped with a Nipype implementation of the "antsBrainExtraction.sh" workflow (from ANTs), using OASI-S30ANTs as target template. Brain tissue segmentation of cerebrospinal fluid (CSF), white-matter (WM), and gray-matter (GM) was performed on the brain-extracted T1w using "fast" [FSL 6.0.5.1:57b01774, RRID:SCR_002823, @fsl_fast]. Volume-based spatial normalization to one standard space (MNI152NLin2009cAsym) was performed through nonlinear registration with antsRegistration (ANTs 2.3.3), using brain-extracted versions of both T1w reference and the T1w template. The following template was selected for spatial normalization: ICBM 152 Nonlinear Asymmetrical template version 2009c [@mni152nlin2009casym, RRID: SCR_008796; TemplateFlow ID: MNI152NLin2009cAsym]. For each of BOLD run per subject, the following preprocessing was performed: First, a reference volume and its skull-stripped version were generated using a custom methodology of fMRIPrep. Head-motion parameters with respect to the BOLD reference (transformation matrices, and 6 corresponding rotation and translation parameters) are estimated before any spatiotemporal filtering using "mcflirt" [FSL 6.0.5.1:57b01774, @mcflirt]. BOLD runs were slice-time corrected to 0.52 s (0.5 of slice acquisition range 0 s to 1.04 s) using "3dTshift" from AFNI [@afni, RRID:SCR_005927]. The BOLD time series (including slice-timing correction when applied) were resampled onto their original, native space by applying the transforms to correct for head-motion. These resampled BOLD time series will be referred to as "preprocessed BOLD." The BOLD reference was then co-registered to the T1w reference using "mri_coreg" (FreeSurfer) followed by "flirt" [FSL 6.0.5.1:57b01774, @flirt] with the boundary-based registration [@bbr] cost-function.

Co-registration was configured with 6 degrees of freedom. Several confounding time series were calculated based on the preprocessed BOLD: framewise displacement (FD), DVARS, and 3 region-wise global signals. FD was computed using 2 formulations following Power (absolute sum of relative motions, @power_fd_dvars) and Jenkinson (relative root mean square displacement between affines, @mcflirt). FD and DVARS are calculated for each functional run,

both using their implementations in Nipype [following the definitions by @power_fd_dvars]. The 3 global signals are extracted within the CSF, the WM, and the whole-brain masks. Additionally, a set of physiological regressors were extracted to allow for component-based noise correction [CompCor, @compcor]. Principal components are estimated after high-pass filtering the preprocessed BOLD time series (using a discrete cosine filter with 128 s cut-off) for the 2 CompCor variants: temporal (tCompCor) and anatomical (aCompCor). tCompCor components are then calculated from the top 2% variable voxels within the brain mask. For aCompCor, 3 probabilistic masks (CSF, WM, and combined CSF+WM) are generated in anatomical space. aCompCor masks are subtracted a mask of pixels that likely contain a volume fraction of GM. This mask is obtained by thresholding the corresponding partial volume map at 0.05, and it ensures components are not extracted from voxels containing a minimal fraction of GM. Finally, these masks are resampled into BOLD space and binarized by thresholding at 0.99 (as in the original implementation). Components are also calculated separately within the WM and CSF masks. For each CompCor decomposition, the k components with the largest singular values are retained, such that the retained components' time series are sufficient to explain 50 percent of variance across the nuisance mask (CSF, WM, combined, or temporal). The remaining components are dropped from consideration. The head-motion estimates calculated in the correction step were also placed within the corresponding confounds file. The confound time series derived from head motion estimates and global signals were expanded with the inclusion of temporal derivatives and quadratic terms for each [@confounds_satterthwaite_2013]. Frames that exceeded a threshold of 0.5 mm FD or 1.5 standardized DVARS were annotated as motion outliers. The BOLD time series were resampled into standard space, generating a preprocessed BOLD run in MNI152NLin2009cAsym space. First, a reference volume and its skull-stripped version were generated using a custom methodology of fMRIPrep. All resamplings can be performed with a single interpolation step by composing all the pertinent transformations (i.e., head-motion transform matrices, susceptibility distortion correction when available, and co-registrations to anatomical and output spaces). Gridded (volumetric) resamplings were performed using "antsApplyTransforms" (ANTs), configured with Lanczos interpolation to minimize the smoothing effects of other kernels [@lanczos]. Non-gridded (surface) resamplings were performed using "mri_vol2surf" (FreeSurfer). Many internal operations of fMRIPrep use Nilearn 0.8.1 [@nilearn, RRID:SCR_001362], mostly within the functional processing workflow. For more details of the pipeline, see https://fmriprep.readthedocs.io/en/latest/workflows.html.

## Univariate analysis

All univariate group-level fMRI analyses were conducted at the whole-brain level and corrected for multiple comparisons using FSL Randomise with 5,000 permutations. Randomise uses a permutation-based statistical inference that does not rely on a Gaussian distribution [74]. A statistical threshold of $p < 0.05$, corrected for multiple comparisons with familywise error (FWE) correction and threshold-free cluster enhancement (TFCE), was used for analyses. TFCE helps identify significant clusters without defining an initial cluster-forming threshold or carrying out a large amount of data smoothing [75]. Whole-brain conjunction analyses were conducted using the easythresh_conj script in FSL [76] and recommended thresholds ($Z > 2.3$, cluster size $p < 0.05$) to identify regions commonly activated for *Safety Prediction* and *Meta-representation*. MRIcron was used for visualization.

## Informational connectivity analysis

Multivariate Informational Connectivity analyses were conducted using the IC Toolbox in Matlab [77]. An advantage of Informational Connectivity over univariate functional

connectivity is that Informational Connectivity utilizes all patterns of responses within regions to code information that is lost by averaging, which identifies functional connections that cannot be found in univariate functional connectivity analyses [23,24]. Furthermore, Informational Connectivity allows us to test regional interactions in terms of specific experimental conditions [24] such as estimating safety in response to threat (animals) versus protection (weapons). Informational Connectivity was measured between every pair of the 10 ROIs (see below "Regions of interest" and Fig 6A). Network-based statistics were calculated for multi-voxel pattern synchronization changes as a function of *Safety Prediction* during first stimulus presentation to partial information (Fig 6B) and *Safety Meta-representation* during second stimulus presentation as a function of paired stimuli (Fig 6C). To identify hubs connecting regions within the networks identified, we computed the betweenness centralities (BCs) of each region, which represents the fraction of all shortest paths that contain a specific node (Fig 6B and 6C). Betweenness Centralities of 0 are not plotted. Connections are reported at $p < 0.05$, corrected based on within-subject permutation testing (10,000 iterations) as per the permute_ROI_IC() function within the IC Toolbox [77].

### Regions of interest (ROIs)

ROIs were used in 2 analyses, multivariate searchlight and multivariate informational connectivity. All other analyses were conducted using the whole brain. ROIs were independently defined based on analytic purposes. First, a large vmPFC mask was independently defined using Neurovault and applied as a small-volume correction for the Safety Value Updating searchlight analysis. The vmPFC mask (https://identifiers.org/neurovault.image:132836) consisted of 152 subjects and 4,233 voxels. The mask was transformed to standard space (MNI152NLin2009cAsym) using "flirt" [FSL 6.0.5.1:57b01774, @flirt]. A large vmPFC mask was used to assess whether different subregions of the vmPFC were implicated in Safety Value Updating rather than constraining search to a specific subregion. Second, 10 separate ROIs were defined for Multivariate Informational Connectivity analyses. Regions were selected based on threat acquisition and extinction circuitry in prior literature and defined using the Harvard-Oxford cortical and subcortical structural atlases, except for the PAG which was defined using Neurosynth meta-analysis (https://neurosynth.org/; "periaqueductal" with an association test). Instead of using the larger homogenous vmPFC structure as constrained during searchlight, we divided the vmPFC into anterior and posterior subparts based on the Harvard-Oxford atlas to examine distinct connectivity within these regions. The vmPFC ROIs used in Informational Connectivity analyses were contained within the vmPFC mask used in the searchlight. The 10 ROIs selected were the anterior vmPFC ("frontal medial cortex"), posterior vmPFC ("subcallosal cortex"), thalamus, ACC, bilateral insula, dorsal striatum, ventral striatum, PAG, bilateral amygdala, and bilateral hippocampus.

### Preregistration

Hypotheses and methods were preregistered on the Open Science Framework (OSF), https://osf.io/hw3r9.

### Supporting information

**S1 Text. Table A in S1 Text.** Mixed effects models predicting safety prediction (probability of choosing "win") from stimuli safety value, split by stimulus order and type. **Table B in S1 Text. Neural response to safety.** Significant clusters from group level whole-brain univariate analyses. **Table C in S1 Text. Neural response to danger**. Significant clusters from group level whole-brain univariate analyses. **Fig A in S1 Text. Figure A. Neural response to danger**

**increases during each task phase, highlighting regions of canonical defensive circuitry involved in threat detection such as the insula, thalamus, and PAG.** All analyses were conducted using FSL Randomise, TFCE, FWE-corrected $p < 0.05$. Color bar indicates t-intensity values. **(Fig A panels A–C)** Parametric increases in whole-brain neural activity that track decrease in experimentally established safety value of stimuli during **Danger Prediction**. The first stimulus presented represented a bias to partial information, which measures a differentiation in neural activity as a function of stimulus type (threat versus protection). Significant clusters indicate activation increased in those regions as safety probability decreased. Safety decrease was based on the average experimentally established safety probability of each stimulus (protection continuum order: fist, stick, gun grenade; threat continuum order: cat, goose, lion, grizzly). (Fig A panel A) Threat and Protection collapsed, (Fig A panel B) Threat only, (Fig A panel C) Protection only. **(Fig A panels D–F)** Parametric increases in whole-brain neural activity that track the increased experimentally established safety value of stimuli during **Danger Meta-representation.** The second stimulus safety value was based on the combined danger probability of the first and second stimuli. For analyses, safety was based on comparison with the average safety value of the stimulus and examined for trials where safety decreased. For example, if a stick was shown as the second stimulus and was paired with a lion, the probability of safety would reduce from 35.72% (safety average for all stick trials) to 21.43% (safety when stick is paired with lion) (see Fig 1B). **(Fig A panel D)** Threat and Protection collapsed, **(Fig A panel E)** Threat only, **(Fig A panel F)** Protection only. **(Fig A panel G)** Neural activation in response to **Danger Recognition** when subjects learned they were unsuccessful in battle. Analyses probed response at the outcome screen when it indicated potential for electric shock (20%) compared to when it indicated certain safety from shock (100%). Source data can be found at https://osf.io/8qg7y/ under "MRI data." **Fig B in S1 Text. Results of stimuli development and selection of stimuli at the high and low ends of the safety estimation spectrum.** Two questions were asked related to level of danger (animals) and power (weapons) of 20 potential stimuli images. Items were also paired in head-to-head battles with all other stimuli of the same type. Lion and grizzly were rated as the most dangerous stimuli and cat and goose were rated as the second and third least dangerous stimuli (rat was selected as the least dangerous but ultimately excluded from the set to avoid conflating threat with disgust). The same rankings were reported for the head-to-head battles across all animals. The grenade and gun were rated as the most powerful weapons and as most likely to win head-to-head. Fist and stick were rated in the bottom 30% of power ratings and bottom 20% of head-to-heads. Other weapons rated as less powerful were excluded due to concerns of unwieldy usage (i.e., rope). Stimulus design was inspired by 2021 YouGov survey of 1,224 adults showing that that 6% of Americans believe they could beat a grizzly bear in a fight without weapons (retrieved from: https://today.yougov.com/society/articles/35852-lions-and-tigers-and-bears-what-animal-would-win-f). Animals were selected as pilot stimuli based on those survey results and a selection of weapons across the range of potential danger was also tested. Animal images were selected to be "attacking" to mitigate any issues of liking the animal (cuddly housecat versus angry housecat) to ensure stimuli were immediately recognizable as a threat. Weapon images were selected to be without any other stimuli in the picture and on white backgrounds. All images were kept in black and white and sepia tone ranges.
(DOCX)

## Author Contributions

**Conceptualization:** Sarah M. Tashjian, Dean Mobbs.

**Data curation:** Sarah M. Tashjian, Wenning Deng.

**Formal analysis:** Sarah M. Tashjian, Joseph Cussen, Wenning Deng, Bo Zhang.

**Funding acquisition:** Sarah M. Tashjian, Dean Mobbs.

**Investigation:** Sarah M. Tashjian.

**Methodology:** Sarah M. Tashjian.

**Project administration:** Sarah M. Tashjian.

**Supervision:** Sarah M. Tashjian, Dean Mobbs.

**Validation:** Sarah M. Tashjian.

**Visualization:** Sarah M. Tashjian, Joseph Cussen.

**Writing – original draft:** Sarah M. Tashjian.

**Writing – review & editing:** Dean Mobbs.

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
