## [Editor Report · Decision Letter 0]

27 Aug 2024

Dear Dr Tashjian, 

Thank you for submitting your manuscript entitled "Adaptive Safety Coding in the Prefrontal Cortex" for consideration as a Research Article by PLOS Biology.

Your manuscript has now been evaluated by the PLOS Biology editorial staff and I am writing to let you know that we would like to send your submission out for external peer review.

Please note that we unfortunately have not been able to receive advice from one of our academic editors on your study and have, therefore, not yet made a firm decision on whether the conceptual advance is sufficient for PLOS Biology. We will discuss this after review with one of our editorial board members and will be looking for strong reviewer support.

Once your full submission is complete, your paper will undergo a series of checks in preparation for peer review. After your manuscript has passed the checks it will be sent out for review. To provide the metadata for your submission, please Login to Editorial Manager (https://www.editorialmanager.com/pbiology) within two working days, i.e. by Aug 29 2024 11:59PM.

Kind regards,

Christian

Christian Schnell, PhD

Senior Editor

PLOS Biology

cschnell@plos.org

---

## [Decision Letter · Decision Letter 1]

17 Oct 2024

Dear Dr Tashjian,

Thank you for your patience while your manuscript "Adaptive Safety Coding in the Prefrontal Cortex" was peer-reviewed at PLOS Biology. It has now been evaluated by the PLOS Biology editors, an Academic Editor with relevant expertise, and by several independent reviewers. 

In light of the reviews, which you will find at the end of this email, we would like to invite you to revise the work to thoroughly address the reviewers' reports.

As you will see below, the reviewers find the topic of your study important and are overall supportive of publication of your study. However, they also raise a few concerns that need to be addressed. Among the concerns raised, we would like to highlight the need to (i) address Reviewer 1's comment to map the different sub-regions of vmPFC to any specific anatomical boundaries and speculate on why these different portions were involved in the different processes, (ii) more carefully define and introduce the concepts and how they relate to your study, and (iii) provide evidence that the task did engage the constructs and processes it is supposed to engage and that the fMRI sequence is suitable.

Given the extent of revision needed, we cannot make a decision about publication until we have seen the revised manuscript and your response to the reviewers' comments. Your revised manuscript is likely to be sent for further evaluation by all or a subset of the reviewers.

**IMPORTANT - SUBMITTING YOUR REVISION**

*Re-submission Checklist*

*Published Peer Review*

*PLOS Data Policy*

*Blot and Gel Data Policy*

Sincerely,

Christian

Christian Schnell, PhD

Senior Editor

PLOS Biology

cschnell@plos.org

REVIEWS:

Reviewer #1 (Patricia Lockwood): This is an exciting, novel and rigorously conducted study that I enjoyed reading. The behavioural and neural mechanisms of safety and protection remain poorly understood but are crucial drivers of human valuation and decision-making. The combination of a behavioural study with an fmri study is impressive, and studies were preregistered. The use of real shock stimuli is challenging to implement but adds ecological validity to the task. The analyses are sophisticated including univariate and multivariate approaches as well as connectivity analyses. The conclusions are balanced and appropriate. I have just a few comments the authors may wish to consider. 

1. It is very interesting to find different sub-regions of vmPFC involved in different aspects of the task. I wondered if the authors could map these to any specific anatomical boundaries and speculate on why these different portions were involved in the different processes. In light of this I would also hesitate to suggest that these portions represent a gradient but rather different portions of vmPFC.

2. I wonder how the authors see their results fitting with the work on self prioritisation and self relevance in vmPFC? (e.g. Sui & Humphreys, 2015; Sui & Gu., 2017; Lockwood et al., 2018). As several studies have shown that self relevant, as opposed to stranger and friend relevant information, is preferentially encoded in vmPFC. 

Minor

It could be helpful for the figure legends to start with a more informative subtitle than 'neural activation' and 'behavioural results'. A brief summary of the main finding of each figure would be helpful for readers to grasp the main take home from the figure as an anchor for the longer legends. 

Reviewer #2: Tashjian and colleagues submit a manuscript describing highly interesting and timely results form a study that combined a novel behavioral paradigm with fMRI. The study specifically examined whether the level of safety (e.g. having a weapon) influences the perception of a threat (e.g. dangerous animal) and further segregates the behavioral and neural subprocesses involved. The study is well designed, the manuscript is well organized, the results address an important gap in the literature and are cautiously interpreted. I have mainly minor points that may help to improve the manuscript further: 

- Authors discuss the background of the study and the results in the context of fear models. I wonder whether the design of the paradigm and the interpretation may also link with the Theory of Emotion from Lazarus, a cognitive emotion theory which underscorese the role of (re)appraisal (primary, secondary) in emotion processes. The level of evaluation that the authors operationalize in their paradigm is on a rather high cognitive evaluation level. 

- The vmPFC has been engaged in several functions related to fear and safety, but also associated phsyiological reactions such as regulation of cardiac reactivity etc. In their account the authors relate this region to meta-representation and updating. Authors may want to discuss or provide further analyses to differentiate the cognitive computations from the associated phsyiological reactivity or regulation. 

- For the multivariate connectivity authors report in the text that results are presented on the level of "all connections p<0.05". Given the number of multivariate connections please clarify if multiple comparisons were applied. 

- Authors report that (line 337) 'subjects fixated more on the initial safety value if protection stimuli were presented first…. ' however, it appears that the analyses in the behavioral and MRI sample were not consistent (p = .004, and p = .75). Please clarify and interpret the potential differences. 

- Authors observed that activation corresponding to decreases in safety was observed in the occipital pole and postcentral gyrus (in Fig. S1a-b). How could this be explained? Do these regions reflect preparatory visual and motor engagement to facilitate threat detection? 

Reviewer #3: Tashjian and colleagues provide a novel approach to unpacking the neural dynamics of safety processes in the human vmPFC, combining behavioral and fMRI experiments to target understudied domains of safety encoding. Their findings appear to fit well with the known role of vmPFC in safety processing but also highlight more precise mechanistic contributions and differentiate discrete roles of vmPFC subregions (anterior vs posterior), and wider multivariate brain connectivity. I am impressed by the sophistication of the task - and the results are compelling. For these to be convincing, additional details on the study design, experiment methods, and analyses that should be provided. Conceptually, there ought to be clearer explanation of certain key concepts that are central to the work (e.g., meta-representation), which may not be familiar to other researchers who work on safety processing and the vmPFC and require greater contextualizing. I hope my comments below are helpful in outlining these points.

Major Comments:

1. The main strength of the work also presents some potential areas of concern. Authors provide some novel and interesting angles to the safety processing literature, which I agree do well to complement existing models (e.g., avoidance/Pavlovian paradigms). However, the high novelty means that these concepts need to be well supported, defined, and justified. For example, phrases like 'meta-representation' may seem obscure to some. First, it is necessary (where applicable) to clarify the extent to which these kinds of constructs are entirely new - introduced here by the authors for the first time - or if they are using concepts that already exist in other research areas and leveraging these to gain a better model of safety-related neural processes. 

2. Overall, the figures are useful, but the legends could be improved to clarify pertinent details. For example, Figure 1's legend should be able to explain each part of the figure - leaving no nomenclature unturned. Terms 'R1', R2, etc., are used, as is 'naïve', in parentheses, for certain areas of the figure. Please ensure there is no room for confusion, as understanding the experiment design in full helps understand the results. Most of the other figure legends have very comprehensive descriptions - which are appreciated.

3. More detailed reporting is required when outlining experiment task parameters, such as the length of the inter-trial/inter-stimulus intervals, and so forth, for all parts of the experiments. It should not be too onerous to add seconds/milliseconds in parentheses wherever applicable in-text for the methods sections. I hasten to note that the authors ought not to place too much reliance in figures or in-text descriptions exclusively, the methods should provide all relevant details, and these can be illustrated in figures as a useful adjunct, but ultimately, critical parameters ought not be missing in-text. Other examples of critical information include stimulus durations, how these varied in sync with behavioral responses (if they did), and the time gaps/breaks between task phases (even if approximate). It is a very novel and idiosyncratic set of tasks, and if one sought to replicate/translate the tasks, these details are critical.

4. As per above, please reconcile/explain using terms ITI and ISI (intertrial and interstimulus intervals) to describe a similar thing in the same paragraph (page 25).

5. The methods for designing stimuli for proper validity appear highly commendable and the detail in reporting is appreciated. Firstly, if there is any available data as to the choice of stimulus design, this could be helpful in supplementary material. Secondly, concerning post-task checks and balances, I would be interested to know if authors assured construct validity of the tasks, in terms of participants having the intended experiences. For instance, on what basis do authors feel confident that they successfully evoked and manipulated thoughts/feelings/behaviors that were selective to the construct of 'safety' versus other affective states (or merely nonaffective processing)? At a basic level, can we be confident in assuming that this study effectively evoked active safety-specific processes, rather than non-emotional/natural stimulus processing/estimations? Finally, the animals and defensive items are likely to have varied in terms of inherent pleasant/unpleasant/anxiety-arousing connotations (e.g., domestic cats compared to charismatic megafauna, or sticks compared to guns/knives). To be clear, I am strongly convinced by behavioral data (Fig2) that the stimuli and task worked well for computing the probabilities of safety/threat. Though not directly central, I wonder if the affective value (prior to any learning) may be an interesting component to prediction/estimation/meta-representation.

6. Related to the above comment, do authors confidently assert that there was/wasn't some element of associative learning occurring/interacting with their intended effects? It is hard to avoid the possibility that partial aversive reinforcement (20% shocked 'lost battle' trials) may have induced learning effects. Since the task seems to emphasize inherent properties of stimuli (e.g., assumed knowledge about the efficacy of handguns vs sticks, or danger of house cats vs lions), is there a potential influence of shock-induced associative fear/safety learning? If this is a large factor, then how much can it truly be said the task differs from 'external' threat/safety tasks (e.g., fear conditioning/extinction).

7. The 'objective' underlying probabilities of threat, concerning the different combinations of animals + defense strategies (e.g., Fig.1C) is an appealing means for ecological validity in the task (combining a bear and gun = higher % of safety versus combination of bear and stick). Since these are 'objective' and represented in discrete numerical values (64.29%, 78.57%, as examples), can authors please clarify their methods for deriving said percentages? This was a little unclear to me on first reading. The term 'objective' may also raise eyebrows. How do we deliberate on the issue of whether a bear + gun situation is objectively more/less safe than a bear + hand grenade scenario? Is the room for variability here a feature or a bug?

8. Details are provide concerning what was performed for each group, but I would appreciate clarification on why. For the task features present/absent in behavioral/fMRI groups, please explain reasons for there being a difference, the purpose of said differences.

9. Briefly, given the high spatial and temporal resolution of the fMRI acquisition (for a 3Tesla scanner, settings seem to have a combination of relatively small voxels + fast TR), please explain confidence in the quality of imaging data. What head coil (e.g., number of channels) was used? Did fmriprep provide useful quality checks that validated basic levels of data quality? Additionally, do the authors consider accelerated resolution of this kind to be a generally applicable approach or are these settings tuned to be sensitive to their specific study needs?

10. I became a little lost looking between the results, methods, and figures concerning how many ROIs were included and which analyses these were each applied to. Based on its current form, the in-text methods suggest 1 vmPFC ROI, but it also there are at least 2 as per the multivariate connectivity results (Fig.4, for example). I believe the use of multivariate searchlight analysis may obscure things. For readers grounded in more conventional fMRI approaches, clarification is needed on which results were whole-brain, ROI, or where regions were defined based on searchlight search within an ROI (or indeed across whole brain if this was done). To be specific, there needs to be clarity on how authors identified one area of voxels as being region X, and another as region Y. There are phrases in some parts of the paper that suggest whole brain searchlight identified the vmPFC, others that indicate that searchlight was more targeted. Wording must remain consistent - as to whether activations in certain areas were discovered incidentally or sought out explicitly.

11. Some areas use mathematical notation or symbols in ways that needlessly complicate simple things. Specifically, conjunction analyses. Figure 3's legend includes "Conjunction analyses for the safety prediction increasing safety > increasing danger ∩ safety meta-representation increasing safety > increasing danger for all stimuli, with overlapping activation in the vmPFC; Z=2.3, p<.05." which may be missing something. It would doubtless be easier to say there was a conjunction in activation/a shared activation in vmPFC between the two contrasts?

12. In Discussion section, the following sentence presents some obstacles to easy interpretation: "This study identifies neural systems involved in safety coding, provides evidence that Safety Prediction evokes dissociable circuits depending on whether the stimulus has self-relevance, and supports the hypothesis that the brain integrates threat and protective information to Meta-represent safety.". As with some other parts of the paper, there is perhaps too much assumed knowledge on the part of the reader as to knowing (for example) what it means 'to meta-represent' something, or what the authors are referring to by 'self-relevance' in the context of their current task. The latter is particularly important since the vmPFC is so closely linked with self-related processes - which are often very different from the authors' task. This comment is not specific to this sentence however, it may only require one area of the manuscript where such concepts or framings are precisely defined.

Minor Comments:

13. Page 3, line 57 "reducing stress, and initiating in other survival" - typo - the 'in' is probably not meant to be there

14. Page 3 line 67: "…relevant information changes so can the safety estimate even if the…" - a comma needs to separate 'estimate' and 'even' I think.

15. In some figures the brain activation images that are surrounded by small grey borders that appear to be unintended, perhaps an issue with converting file types. It should be fixed in next submission. Likewise, Figure 3 panel J seems to be showing some errors, half the axial slice is cut off (assuming this is not intentional).

---

## [Editor Report · Decision Letter 2]

11 Dec 2024

Dear Dr Tashjian,

Thank you for your patience while we considered your revised manuscript "Adaptive Safety Coding in the Prefrontal Cortex" for publication as a Research Article at PLOS Biology. This revised version of your manuscript has been evaluated by the PLOS Biology editors and the Academic Editor.

Based on our Academic Editor's assessment of your revision, we are likely to accept this manuscript for publication, provided you satisfactorily address the address the following data and other policy-related requests.

* We would like to suggest a different title to improve its accessibility for our broad audience: "Subregions in the ventromedial prefrontal cortex integrate threat and protective information to meta-represent safety"

* Please include the approval/licences number of the ethical approval from the institutional review board.

* Please include information in the Methods section whether the study has been conducted according to the principles expressed in the Declaration of Helsinki.

* Please specify whether the participants provided written or oral consent.

* DATA POLICY:

Regardless of the method selected, please ensure that you provide the individual numerical values that underlie the summary data displayed in the following figure panels as they are essential for readers to assess your analysis and to reproduce it: 2A, 4BC and S2.

* CODE POLICY

* Please note that per journal policy, the model system/species studied should be clearly stated in the abstract of your manuscript. 

We expect to receive your revised manuscript within two weeks. 

*Published Peer Review History*

*Press*

Sincerely,

Christian

Christian Schnell, PhD

Senior Editor

cschnell@plos.org

PLOS Biology

---

## [Editor Report · Decision Letter 3]

16 Dec 2024

Dear Dr Tashjian,

Thank you for the submission of your revised Research Article "Subregions in the ventromedial prefrontal cortex integrate threat and protective information to meta-represent safety" for publication in PLOS Biology. On behalf of my colleagues and the Academic Editor, Benjamin Becker, I am pleased to say that we can in principle accept your manuscript for publication, provided you address any remaining formatting and reporting issues. These will be detailed in an email you should receive within 2-3 business days from our colleagues in the journal operations team; no action is required from you until then. Please note that we will not be able to formally accept your manuscript and schedule it for publication until you have completed any requested changes.

While you attend to the requests to come, please also add a statement to the corresponding figure legends where the source data can be found. For example: "Source data can be found https://osf.io/8qg7y/." Please also specify the folder, so it's easier for readers to find the corresponding data.

PRESS

Sincerely, 

Christian

Christian Schnell, PhD

Senior Editor

PLOS Biology

cschnell@plos.org